# In Vitro Expansion of Vδ1+ T Cells from Cord Blood by Using Artificial Antigen-Presenting Cells and Anti-CD3 Antibody

**DOI:** 10.3390/vaccines11020406

**Published:** 2023-02-10

**Authors:** Gaeun Hur, Haeyoun Choi, Yunkyeong Lee, Hyun-Jung Sohn, Su-Yeon Kim, Tai-Gyu Kim

**Affiliations:** 1Department of Microbiology, College of Medicine, Catholic University of Korea, Seoul 06591, Republic of Korea; 2Department of Biomedicine & Health Sciences, College of Medicine, Catholic University of Korea, Seoul 06591, Republic of Korea; 3Catholic Hematopoietic Stem Cell Bank, College of Medicine, Catholic University of Korea, Seoul 06591, Republic of Korea

**Keywords:** γδ T cells, cord blood, Vδ1+ T cells, PBMC, anti-CD3 antibody, artificial antigen presenting cells

## Abstract

γδ T cells have the potential for adoptive immunotherapy since they respond to bacteria, viruses, and tumors. However, these cells represent a small fraction of the peripheral T-cell pool and require activation and proliferation for clinical benefits. In cord blood, there are some γδ T cells, which exhibit a naïve phenotype, and mostly include Vδ1+ T cells. In this study, we investigated the effect of CD3 signaling on cord blood γδ T-cell proliferation using K562-based artificial antigen presenting cells expressing costimulatory molecules. There were significantly more Vδ1+ T cells in the group stimulated with anti-CD3 antibody than in the group without. In cultured Vδ1+ T cells, DNAM-1 and NKG2D were highly expressed, but NKp30 and NKp44 showed low expression. Among various target cells, Vδ1+ T cells showed the highest cytotoxicity against U937 cells, but Daudi and Raji cells were not susceptible to Vδ1+ T cells. The major cytokines secreted by Vδ1+ T cells responding to U937 cells were Granzyme B, IFN-γ, and sFasL. Cytotoxicity by Vδ1+ T cells correlated with the expression level of PVR and Nectin of DNAM-1 ligands on the surface of target cells. Compared to Vδ2+ T cells in peripheral blood, cord blood Vδ1+ T cells showed varying cytotoxicity patterns depending on the target cells. Here, we determined the ideal conditions for culturing cord blood Vδ1+ T cells by observing that Vδ1+ T cells were more sensitive to CD3 signals than other subtypes of γδ T cells in cord blood. Cultured cord blood Vδ1+ T cells recognized target cells through activating receptors and secreted numerous cytotoxic cytokines. These results are useful for the development of tumor immunotherapy based on γδ T cells.

## 1. Introduction

γδ T cells express distinct T-cell receptor (TCR) γ and δ chains on their surface, accounting for 0.5%–5% of all T-lymphocytes in peripheral blood [1]. Particularly, they are more widespread within epithelial tissues where they can comprise up to 50% of T cells [2]. Unlike αβ T cells, these cells are not restricted by the major histocompatibility complex and could respond to antigens without conventional antigen processing [3,4,5,6]. They rapidly increase in patients with infectious diseases and tumors, producing various cytokines according to inflammatory environments [5,7,8,9]. The Vδ2+ and non-Vδ2+ (mainly Vδ1+) subsets of human γδ T cells have potent antitumor effects due to distinct homing patterns and activating killer receptors [10]. While the TCR of Vδ2+ T cells primarily recognizes small bacterial phosphoantigens, alkylamines, and tumor cell-derived pyrophosphates, Vδ1+ T cells preferentially recognize stress-inducible ligands such as MICA/B, ULBPs, and several other ligands [11,12].

Recently, γδ T cells have been applied for adoptive tumor immunotherapy. However, they require activation and propagation for clinical benefits as they are only a small fraction of the peripheral T-cell pool [13]. A Vδ2+ subset of γδ T cells expanded with aminobisphosphonates has been tested in clinical trials of cancer [14,15]. Meanwhile, agonistic antibodies stimulating gd T cells and artificial antigen presenting cells (aAPCs) have been successfully used in expansion protocols for non-Vδ2+ T cells [16]. Meanwhile, for applicable expansion protocols for non-Vδ2+ T cells, agonistic antibodies stimulating γδ T cells, artificial antigen presenting cells (aAPCs), have been successfully used in expanding γδ T cells [16]. For clinical trials, sufficient γδ T cells have been produced using aAPCs, showing tumor reactivity in vitro and in vivo [13]. Moreover, a clinical-grade culture method of Vδ1+ T cells devoid of feeder cells was developed for adoptive immunotherapy [17].

Cord blood is whole blood obtained from the umbilical cord at childbirth and is an alternative source of stem cells useful for hematopoietic stem cell transplantation [18]. Most γδ T cells in adult peripheral blood express Vδ2 (>70%) [19,20], but the proportion can change in certain situations [21,22]. In cord blood, γδ T cells are present at a low frequency (<1%) and express a naïve phenotype [19,23]. The repertoire is polyclonal and most express Vδ1; Vδ2 is usually present in ≤25% [24,25]. In vitro cord blood γδ T-cell expansion has several limitations, including a low number of γδ T cells, low proportion of Vδ2+ T cells responding to phosphoantigens, and their immature phenotype [19,26]. Even if culture conditions of cord blood γδ T cells based on zoledronic acid and IL-2 showed proliferation of Vδ2+ T cells, Vδ1+ T cells showed relatively low proliferation and a low distribution, approx. 10% [19]. When γδ T cells isolated from cord blood were stimulated with K562-based aAPCs, polyclonal TCR γδ T cells proliferated [27]. Previously, we reported that γδ T cells from adult peripheral blood can be cultured for a long time using K562 cells expressing CD80, CD83, and CD137L [28]. In addition, γδ T cells cultured in this way showed a higher antitumor effect than γδ T cells cultured using zoledronic acid in preclinical animal models [29]. Here, we further investigated the effect of anti-CD3 antibody stimulation on cord blood γδ T-cell proliferation using these aAPCs, finding that, compared to other subtypes, CD3 signals are required for the proliferation of Vδ1+ T cells. Vδ1+ T cells cultured in cord blood showed different cytotoxicity to various target cells compared to Vδ2+ T cells cultured in peripheral blood.

## 2. Materials and Methods

### 2.1. Cord Blood and Peripheral Bloods and Cell Lines

Human cord blood and peripheral blood were provided by the Catholic Hematopoietic Stem Cell Bank. The use of human material was reviewed and approved by the Institutional Review Board of the College of Medicine, Catholic University of Korea, Seoul, Republic of Korea (permit No. MC18SESI0111). All subjects provided written informed consent for sample donation in accordance with the Declaration of Helsinki.

K562, U937, Daudi, and Raji cells were obtained from the American Type Culture Collection (Manassas, VA, USA). aAPCs in which costimulatory molecules (CD32, CD80, CD83, 4–1BBL, CD40L, and CD70) were stably expressed in K562 cells [30], using a lentiviral vector, were used to stimulate and proliferate γδ T cells in culture. Cell culture was maintained in RPMI 1640 medium (Lonza, Walkersville, MD, USA) supplemented with 10% fetal bovine serum (Gibco, Grand Island, NY, USA), 1% L-glutamine (Lonza, Walkersville, MD, USA), and 1% penicillin/streptomycin (Lonza, Walkersville, MD, USA) in an incubator at 37 °C, 95% humidity, and 5% CO_2_.

### 2.2. Cord Blood γδ T-Cell Expansion

Cord blood mononuclear cells were isolated by Ficoll-Hypaque (GE Healthcare, Pittsburgh, PA, USA) density gradient centrifugation. γδ T cells were negatively isolated using magnetic microbeads (MACS, Miltenyi Biotec, Bergisch Gladbach, Germany) according to the manufacturer’s instructions. RPMI 1640 medium (Lonza, Walkersville, MD, USA) supplemented with 10% fetal bovine serum (Gibco, Grand Island, NY, USA), 1% L-glutamine (Lonza, Walkersville, MD, USA), and 1% penicillin/streptomycin (Lonza, Walkersville, MD, USA) was used as a basic culture medium for γδ T cells. γδ T cells, at 5 × 10^5^ cells/mL, were co-cultured in a 24-well plate with irradiated (100 Gy) aAPCs at a 2:1 T cell:aAPCs ratio with 200 U/mL of IL-2 (Proleukin; Novartis, East Hanover, NJ, USA) with/without 500 ng/mL of anti-CD3 antibody (clone OKT3, #16–0037-85, Invitrogen, Waltham, MA, USA) and added with 200 U/mL of IL-2 (Proleukin; Novartis, East Hanover, NJ, USA) on day 3. To culture total γδ T cells, cells were further cultured under the same conditions on day 7, refreshed with 200 U/mL of IL-2 in complete media (Proleukin; Novartis, East Hanover, NJ, USA) on day 10, and cultured until day 14.

To culture pure Vδ1+ T cells, cells were labeled with an anti-TCRVδ1-APC (TS8.2) antibody on day 7, and analyzed and sorted by a fluorescence-activated cell sorter FACS Aria Fusion (BD Biosciences, San Diego) using FACSDiva software (BD Biosciences). Vδ1+ T cells, 5 × 10^5^ cells/mL, were co-cultured in a 24-well plate with irradiated (100 Gy) aAPCs at a 2:1 T cell:aAPCs ratio in the presence of 200 U/mL of IL-2 (Proleukin; Novartis, East Hanover, NJ, USA) with 500 ng/mL of anti-CD3 antibody (clone OKT3, #16–0037-85, Invitrogen). On day 10, the media including 200 U/mL of IL-2 (Proleukin; Novartis, East Hanover, NJ, USA) were refreshed, and cells were cultured until day 14. To calculate fold expansion, after sorting on day 7, seeding of 5 × 10^5^ cells per mL was cultured for 1 week; the expanded fold compared to 7 days was calculated.

### 2.3. Vγ9Vδ2 T-Cell Expansion

γδ T cells were maintained in RPMI 1640 medium (Lonza, Walkersville, MD, USA) supplemented with 10% fetal bovine serum (Gibco, Grand Island, NY, USA), 1% L-glutamine (Lonza, Walkersville, MD, USA), and 1% penicillin/streptomycin (Lonza, Walkersville, MD, USA). On day 0, PBMCs were stimulated with 3 μM of zoledronic acid (Daewoong Pharmaceutical Co, Ltd., Seoul, Korea) in the presence of 1000 U/mL of IL-2 (Proleukin; Novartis, East Hanover, NJ, USA) and seeded at 5 × 10^5^ cells/mL in a 24-well plate. From day 7, γδ T cells were treated with IL-2 (1000 U/mL) every 3–4 days, stimulated weekly with irradiated (100 Gy) aAPCs at a 2:1 T:aAPC ratio. On day 21, cells were frozen using a controlled-rate freezer (Thermo Fisher Scientific, Waltham, MA, USA) and cryopreserved in liquid nitrogen until further use. For cytotoxicity assays, frozen γδ T cells were thawed and cultured overnight in the presence of IL-2 (1000 U/mL).

### 2.4. Flow Cytometer Assay

γδ T-cell analyses were performed on day 0, day 7, and day 14. Cells were harvested and stained with the following antibodies: anti-TCRVδ2-FITC (B6), anti-CD45RA-PE (HI100), anti-CD27-APC (M-T271), anti-CD226-Brilliant Violet 421 (11A8), anti-TIGIT-APC (A15153G), anti-CD152-Brilliant Violet 421 (BNI3), anti-CD314 (NKG2D)-APC/Cyanine 7 (1D11), anti-CD336 (NKp44)-PE/Cyanine 7 (P44–8), and anti-CD337 (NKp30)-PerCP/Cyanine5.5 (P30–15), purchased from BioLegend (San Diego, CA, USA); and anti-CD279-PE (MIH4), anti-CD274-APC-eFluor 780 (MIH1), anti-TCRVδ1-APC (TS8.2), and anti-CD45-PerCP-eFluor 710 (HI30), purchased from Invitrogen. Acquisition was performed on a FACSCANTO flow cytometry instrument (BD Biosciences, San Diego, CA, USA) using the FACSDiva software. Acquired data were analyzed with FlowJo (Tree star Inc., Ashland, OR, USA).

### 2.5. Cytotoxicity Assay

For the cytotoxicity assay using calcein AM (Invitrogen), target cells were stained with 5 μM calcein AM and incubated for 20 min at 37 °C and 5% CO_2_ in the dark and washed twice with phosphate-buffered saline and suspended in 10% RPMI at 1 × 105 cells/mL. Effector and target cells were co-cultured at the indicated effector:target ratios in a 96-well U-bottom plate. Triton X-100 was added to a final concentration of 2% for maximum fluorescence release and only target cells were cultured without effector cells for spontaneous release. After a 4 h incubation, plates were centrifuged at 2500 rpm for 5 min, and the supernatants carefully transferred to a black opaque plate. The fluorescence was measured with a Synergy H1 microplate reader (BioTek, Winooski, VT, USA). The specific release was calculated as (Experimental Fluorescence) − (Spontaneous Fluorescence) / (Maximum Fluorescence) − (Spontaneous Fluorescence) × 100.

### 2.6. Secretome Analysis

For secretome analysis, Vδ1+ γδ T cells and Daudi, U937 cells were co-cultured in a 96-well U-bottom plate at a 2:1 ratio for 4 or 24 h; plates were centrifuged at 2500 rpm for 5 min, and the supernatants were carefully transferred to 5 mL round tubes. The supernatants were analyzed using a LEGENDplex human CD8/NK panel (BioLegend) according to the manufacturer’s protocol.

### 2.7. Statistics

Statistical significance was determined by paired *t*-tests, and one-way and two-way analysis of variance. All analyses were performed using Prism 5.01 software (GraphPad Software, San Diego, CA, USA).

## 3. Results

### 3.1. Effect of Anti-CD3 Antibody Stimulation on γδ T-Cell Subtypes’ Proliferation

To investigate the effect of CD3 signals on the γδ T-cell subtypes of cord blood, γδ T cells negatively isolated with magnetic beads were stimulated for two weeks using irradiated K562-based aAPCs expressing costimulatory molecules, in the presence or absence of anti-CD3 antibody, respectively. The subtypes of γδ T cells negatively isolated from 31 cord blood samples were 11.2% ± 12.5% of Vδ1+ γδ T cells, 2.6% ± 2.3% of Vδ2+ γδ T cells, and 86.1% ± 14% of Vδ1-Vδ2- γδ T cells (Figure 1a). A representative plot of distribution of Vδ1, Vδ2, and Vδ1-Vδ2- γδ T cells after negative isolation with magnetic beads from cord blood samples is presented in Appendix A. On day 14, the fold expansion of the group stimulated with anti-CD3 antibody (328.6 ± 226.6) was significantly higher than that of the group without anti-CD3 antibody (23.0 ± 22.5; Figure 1b). In comparison, the proportion of Vδ1+ T cells increased in the group with anti-CD3 antibody than in that without (52.3% ± 5% vs. 29.9% ± 11.9%; Figure 1c). A representative plot of distribution of Vδ1+, Vδ2+, and Vδ1-Vδ2- γδ T cells is presented in Appendix A. The proportions of Vδ2+ T cells and Vδ1-δ2- T cells showed no significant differences between groups. These data suggest that CD3 stimulation is required for γδ T-cell proliferation in cord blood, and that Vδ1+ T cells are the most sensitive to this type of stimulation. Therefore, in the subsequent experiment, we established a culture method to expand pure Vδ1+ T cells.

### 3.2. Characteristics of Selectively Expanded Vδ1+ T Cells

The isolation and culture of pure Vδ1+ T cells isolated directly from cord blood were not efficient due to low cell recovery and viability. Therefore, after culturing negatively isolated γδ T cells with aAPCs and anti-CD3 antibody for one week, Vδ1+ T cells were separated using a FACS cell sorter and cultured for one more week. The proportions of Vδ1+ T cells before and after sorting on day 7 were 12.2% ± 9.4% and 90.6% ± 8%, respectively. After one week of additional culture, Vδ1+ T cells accounted for 93.2% ± 4.4% of all cells (Figure 2a,b). The fold expansion of purely isolated Vδ1+ T cells between days 7 and 14 was 15.97 ± 8 (Figure 2c).

After sorting on the 7th day, about 93% of Vδ1+ T cells were harvested on day 14. To investigate the characteristics of these expanded Vδ1+ T cells, both the differentiation status and surface expression of relevant molecules were investigated. The differentiation markers of Vδ1+ T cells harvested on day 14 were analyzed using flow cytometry. Most cultured Vδ1+ T cells had differentiated into central memory cells (TCM, CD45RA-CD27+, 54.4% ± 18.4%) and effector memory cells (TEM, CD45RA-CD27−, 37.4% ± 22.2%). There were very low frequencies of naïve (CD45RA+CD27+, 0.1% ± 0.2%) and CD45RA+ terminal effector memory cells (TEMRA, CD45RA+CD27−, 0.5% ± 0.9%) Vδ1+ T cells (Figure 3a).

In addition, the levels of activating receptors and inhibitory molecules expressed on the surface of Vδ1+ T cells were analyzed using flow cytometry (*n* = 8). Among the activating receptors, DNAM-1 (97.7% ± 2.1%) and NKG2D (53.3% ± 11.1%) showed high expression whereas NKp30 (5.1% ± 7.4%) and NKp44 (0.2% ± 0.1%) showed low expression. The expression of inhibitory molecules such as PD-1 (8.3% ± 4.5%), CTLA-4 (7.9% ± 11.6%), and PD-L1 (0.1% ± 0.1%) were also very low (Figure 3b). It is rather uncommon that anti-CD3 antibody stimulated T cells do not express PD-1. However, PD-1 expression differs among CD8 T-cell differentiation subsets [31]. In this study, the amplified Vδ1+ T cells mostly had low PD-1 expression as they had a central memory (TCM, CD45RA-CD27+).

### 3.3. Effector Functions of Vδ1+ T Cells

Next, we measured the cytotoxicity of Vδ1+ T cells against various tumor cells and the association with the surface ligands of these target cells. Among U937, K562, Daudi, and Raji cells, U937 cells showed the highest susceptibility to killing by Vδ1+ T cells, whereas K562 cells showed moderate susceptibility. However, Daudi and Raji cells were not susceptible to Vδ1+ T cells. For an E:T ratio of 10:1, the lysis percentage for each target cell was 92% ± 11% for U937 cells, 31.3% ± 23.7% for K562 cells, 4.9% ± 5.3% for Daudi cells, and 1.1% ± 4.4% for Raji cells. To determine the reason for the differing lysis degree among cells, we measured the expression of activating receptor ligands in target cells. In U937 cells, most ligands except MICA were highly expressed, and in K562 cells Nectin and B7-H6 were highly expressed. However, Raji cells showed moderate B7-H6 expression and Daudi cells did not express any ligands. Correlation analysis between the expression rates of each ligand and the cytotoxicity at 10:1 in target cells showed a significant correlation with the expression of PVR (r = 0.95, *p* = 0.041), and with PVR and Nectin of DNAM-1 ligands (r = 0.96, *p* = 0.032) (Appendix A). These results suggest that Vδ1+ T-cell cytotoxicity is affected by the expression levels of activating receptor ligands on the surface of target cells (Figure 4a,b). Representative flow cytometry data of target cell ligands expression is shown in Appendix A.

After co-culturing Vδ1+ T cells and target cells at an E:T ratio of 2:1 for 24 h, supernatants were harvested, and the molecules and cytokines secreted by Vδ1+ T cells were measured by a bead array. When co-culturing Vδ1+ T and U937 cells, the concentrations of Granzyme B, IFN-γ, and sFasL were significantly higher than those in Daudi cells (Figure 5).

Additionally, we compared the cytotoxicity of cord blood Vδ1+ T cells and adult Vδ2+ T cells to U937, K562, and Daudi cells (Figure 6). Vδ2+ T cells were mainly effector memory (73.0 ± 23.3%), and NKG2D (99.6 ± 0.3%) and DNAM-1 (95.8 ± 6.4%) were highly expressed [29]. U937 cells showed significantly higher susceptibility to killing by cord blood Vδ1+ T cells (77.7 ± 10.54%, E:T ratio = 10:1) than adult Vδ2+ T cells (63.8% ± 12.8%, E:T ratio = 10:1). Meanwhile, K562 cells showed lower susceptibility to killing by cord blood Vδ1+ T cells (49.5% ± 17.4%, E:T ratio = 10:1) than adult Vδ2+ T cells (57.2% ± 34%, E:T ratio = 10:1), and Daudi cells showed susceptibility only to adult Vδ2+ T cells (87.5% ± 11.6%, 6% ± 6.9%, E:T ratio = 10:1). These results suggest a different cytotoxicity mechanism of cord blood Vδ1+ T cells to that of adult Vδ2+ T cells.

## 4. Discussion

Although αβ and γδ TCRs both require the CD3 complex for signaling and show similar structures, these receptors had different characteristics in terms of complex geometry, glycosylation pattern, plasma membrane organization, and the accessibility of signaling motifs in the CD3 intracellular tails [32]. In the αβ TCR, an antigen induced conformational change at CD3 subunits is required for αβ TCR activation but not in the human Vδ2+ TCR [33]. Stimulation of CD4/CD8-depleted T cells with OKT3 and γδ TCR monoclonal antibody resulted in the expansion of Vδ2+ T and Vδ1+ T cells (Vδ2 > Vδ1) [13,34]. In this study, we cultured cord blood γδ T cells using aAPCs with or without anti-CD3 antibody, and observed that γδ T cells cultured with anti-CD3 antibody showed higher proliferation than γδ T cells cultured without anti-CD3 antibody; particularly, Vδ1+ T cells proliferated more than Vδ2+ T cells (Figure 1). Stimulation with aAPCs expressing various costimulatory molecules and anti-CD3 antibody may inhibit the apoptosis triggered by TCR/CD3 signaling, inducing proliferation [35].

Cord blood Vδ1+ T cells showed high cytotoxicity in K562 and U937 cells, but not in Raji or Daudi cells. The anti-γδ TCR antibody could expand both the Vδ1 and Vδ2 subsets of peripheral blood γδ T cells. In the comparison of the cytotoxic activity between them, purified Vδ1 and Vδ2 subsets exhibited similar cytotoxicity against Daudi and K562 cells [16]. However, a Vδ1+ T cell line established from human peripheral blood by immortalization with Herpesvirus saimiri was able to specifically recognize K562 cells, but Daudi cells were resistant to Vδ1+ T cell killing [36]. As such, peripheral blood-derived Vδ1+ T cells show different degrees of cytotoxicity to Daudi cells depending on the culture conditions, so it is necessary to confirm this in the future using Vδ1+ T cells amplified under the same culture conditions as in this study. DNAM-1 and NKG2D receptor expression was noted in the Vδ1+ T cells cultured in this study; the expression of PVR, Nectin, ULBPs, and MICA was very low in Raji and Daudi cells compared to K562 or U937 cells. Although B7-H6, an NKp30 ligand, was also highly expressed in K562 and U937 cells, NKp30 expression was very low in cultured Vδ1+ T cells; therefore, it does not seem to play a major role in this type of cytotoxicity. A clinical-grade protocol for expansion and differentiation of “Delta One T” (DOT) cells as peripheral blood Vδ1+ T cells consists of two-steps. In the 1st step, in addition to anti-CD3 antibody, IL-4, IFN-r, IL- 21, IL-1b, and in the 2nd step, a combination of cytokines such as IL-15 and IFN-r was used, and these cytokines increased the proliferation of Vδ1+ T cells and the expression of NKp30 [17]. These DOT cells expressed the natural cytotoxicity receptors, NKp30 and NKp44, which synergized with the T-cell receptor to mediate leukemia cell targeting in vitro. DOT cells showed highly cytotoxic activity against acute myeloid leukemia (AML) primary samples and cell lines. AML reactivity was only slightly impaired upon Vδ1+ TCR antibody blockade, whereas it was strongly dependent on expression of the NKp30 ligand, B7-H6 [37]. Peripheral blood Vδ1+ T cells cultured in the presence or absence of IL-2, soluble anti-CD3 antibody (clone OKT3), and IL-15, showed the highest NKp30 expression (28%) in Vδ1+ T cells cultured with the OKT3/IL-2/IL-15 combination [38]. However, cord blood Vδ1+ T cells showed very low NKp30 and NKp44 expression, which is presumed to be due to the culture using only anti-CD3 antibody and IL-2 in this study. It is necessary to investigate the effects of various cytokines in detail in the future. Since we only used IL-2 in this study, we need to investigate the effects of various cytokines, including IL-15, on the expression of NKp30 or NKp44 receptors [38].

Daudi cells showed the greatest difference in cytotoxicity of cord blood Vδ1+ T cells proliferated compared to adult peripheral blood Vδ2+ T cells. Vδ2+ T cells are activated by cells that accumulate HMBPP and/or IPP and can also be stimulated through receptors NKG2D and DNAM-1 of various stress-induced molecules expressed on tumor cells including MICA/B, UL16-binding proteins (ULBP1–6), Nectin-2, and PVR, respectively [39,40]. Daudi cells generate endogenous phosphorylated metabolites by upregulating the mevalonate pathway, which results in intracellular IPP accumulation [39,41]. The expansion and pattern of responses of multiple Vδ2+ T cells to Daudi cells was similar to that of IPP stimulation [42]. Since PVR, Nectin, ULBPs, MICA, and B7-H6, ligands of the activating receptor of Vδ1+ T cells, were barely expressed in Daudi cells, no Vδ1+ T-cell cytotoxicity was observed. The cytotoxicity of Vδ1+ T cells to U937 cells was significantly higher than that of Vδ2+ T cells; this is probably due to the high expression of the active receptor in Vδ1+ T cells, or differences in the secretion ability of cytokines associated with cytotoxicity (Figure 6). As a result of a blocking test for these receptors using antibodies against DNAM-1 and NKG2D expressed in Vδ1+ T cells, it was confirmed that some of the cytotoxicity of Vδ1+ T cells against U937 cells was partially reduced (Appendix A).

In a co-culture of Vδ1+ T cells and U937, the concentrations of Granzyme B, IFN-γ and sFasL were significantly higher than those of Daudi cells (Figure 5). Perforin and granzymes synergize to mediate target cell apoptosis: pro-apoptotic granzymes diffuse through perforin pores on the plasma membrane of target cells [43]. Vδ1+ T cells showed high cytotoxicity against U937 cells, probably due to the high secretion of both perforin and Granzyme B. However, in a similar test using adult Vδ2+ T cells, IFN-γ, TNF-α, and sFasL, among secreted cytokines after co-culture with Daudi cells, target cells that showed high cytotoxicity were significantly higher than those of U937 with low cytotoxicity [29]. Interestingly, the concentration of Granzyme B secreted from Vδ1+ T cells was about 10 times higher than that of Vδ2+ T cells, while IFN-γ and TNF-α concentrations secreted by Vδ1+ T cells were lower than those of Vδ2+ T cells [29].

In summary, we established the conditions for culturing cord blood Vδ1+ T cells by observing that, among cord blood γδ T cells, Vδ1+ T cells were more sensitive to CD3 signals than other subtypes, and thus effectively proliferated. Cultured cord blood Vδ1+ T cells expressed high DNAM-1 and NKG2D, and showed strong cytotoxicity and high secretion of Granzyme B, sFasL, and IFN-γ by recognizing target cells with their ligands. Although Vδ2+ T cells derived from adult blood were compared with cord blood Vδ1+ T cells in this study, a direct comparison with Vδ1+ T cells derived from adult blood or Vδ1+ derived from other tissue sources is necessary to investigate the biology of human γδ T cells. Since the stimulation of Vδ1+ T cells with aAPC and anti-CD3 antibody used in this study was the same as the αβ T cell stimulation conditions, and the proliferation rate was similar, further study is needed to prove whether the structural properties of the Vδ1 TCR and CD3 complex would have similar structural properties to those of the αβ TCR.

## Figures and Tables

**Figure 1 vaccines-11-00406-f001:**
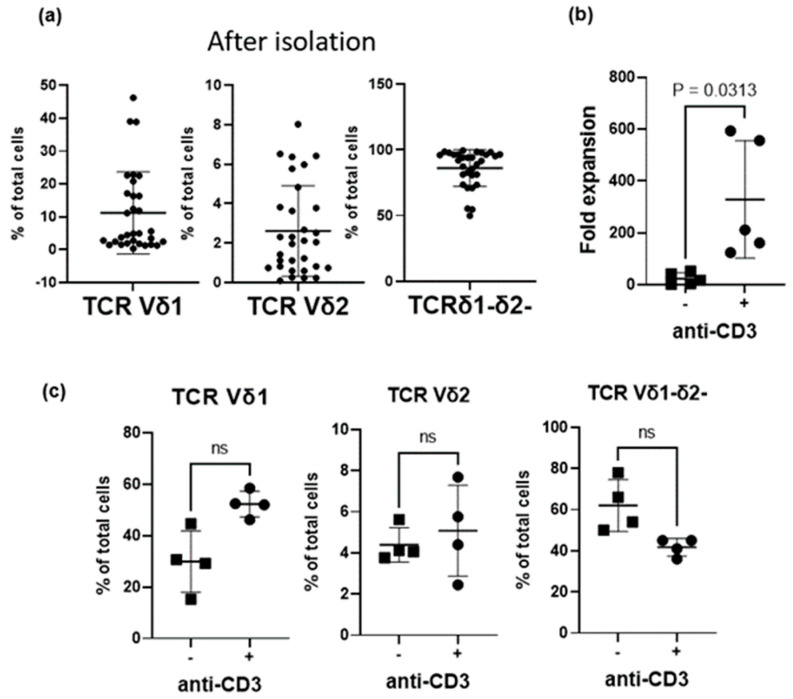
Effects of aAPCs with anti-CD3 antibody on proliferation of γδ T cells negatively isolated from cord blood. (**a**) Distribution of Vδ1, Vδ2, and Vδ1-Vδ2- γδ T cells after negative isolation with magnetic beads from cord blood samples (*n* = 31). (**b**) Increased γδ T-cell proliferation as shown with an anti-CD3 antibody (*n* = 5). (**c**) Distribution of Vδ1+, Vδ2+, and Vδ1-Vδ2- γδ T cells according to the presence or absence of anti-CD3 antibody in culture for two weeks (*n* = 4). Data shown in (**b**,**c**) are the mean ± SD of Vδ1+ T cell frequencies from 4 independent cord blood donors. *p* values were calculated using Wilcoxon matched-pairs signed rank test. ns: not significant, aAPCs: artificial antigen presenting cells.

**Figure 2 vaccines-11-00406-f002:**
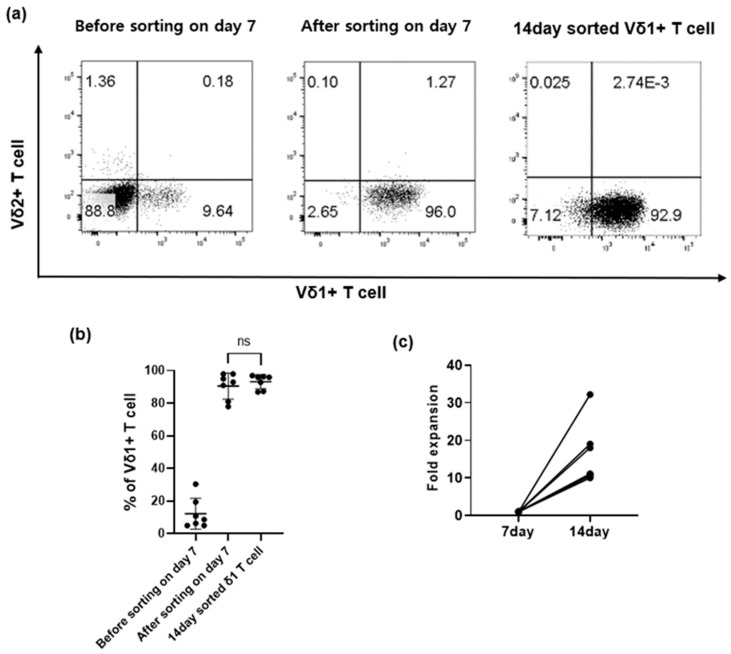
Selective expansion of Vδ1+ T cells from cord blood. (**a**) Flow cytometric analysis of Vδ1+ T cells before and after FACS sorting on the 7th day of culture and after 14 days of culture. (**b**) Proportion of Vδ1+ T cells before and after sorting on day 7 and on day 14 (*n* = 7). (**c**) Fold expansion of Vδ1+ T cells sorted on the 7th day (*n* = 7). Data shown in (**a**) are representative of Vd T cell frequencies from one cord blood donor. Data shown in (**b**,**c**) are the mean ± SD of Vδ1+ T cell frequencies from 7 independent cord blood donors. *p* values were calculated using Wilcoxon matched-pairs signed rank test. ns: not significant.

**Figure 3 vaccines-11-00406-f003:**
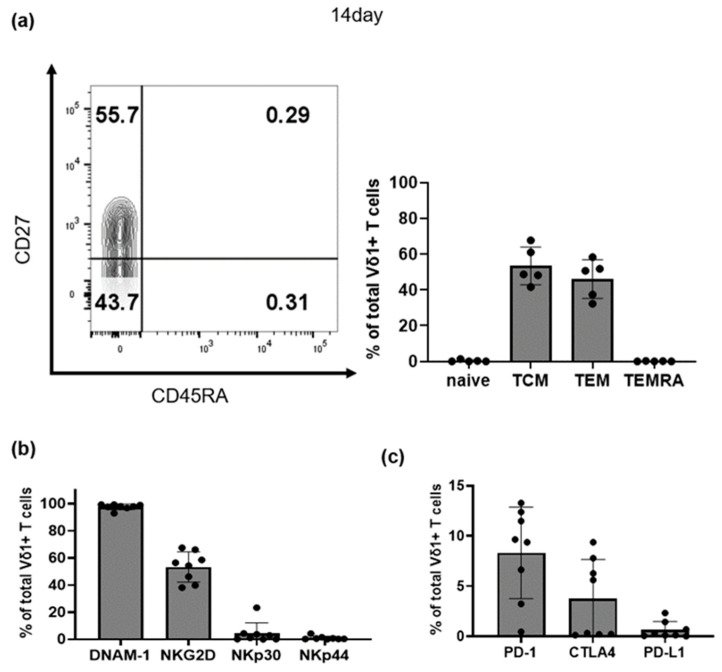
Immunologic memory and cytolytic markers on Vδ1+ T cells. (**a**) Representative flow cytometry plots showing CD27 and CD45RA expression with gates indicating Naïve (CD45RA+CD27+), central memory (TCM, CD45RA-CD27+), effector memory (TEM, CD45RA-CD27−), and CD45RA+ terminal effector memory cells (TEMRA, CD45RA+CD27−) Vδ1+ T cells on day 14. Frequency of TEMRA, effector memory, central memory, and naïve Vδ1+ T cells shown as a percentage of all Vδ1+ T cells (*n* = 5). (**b**) Expression of activating receptors and (**c**) inhibitory molecules on Vδ1+ T cells (*n* = 8).

**Figure 4 vaccines-11-00406-f004:**
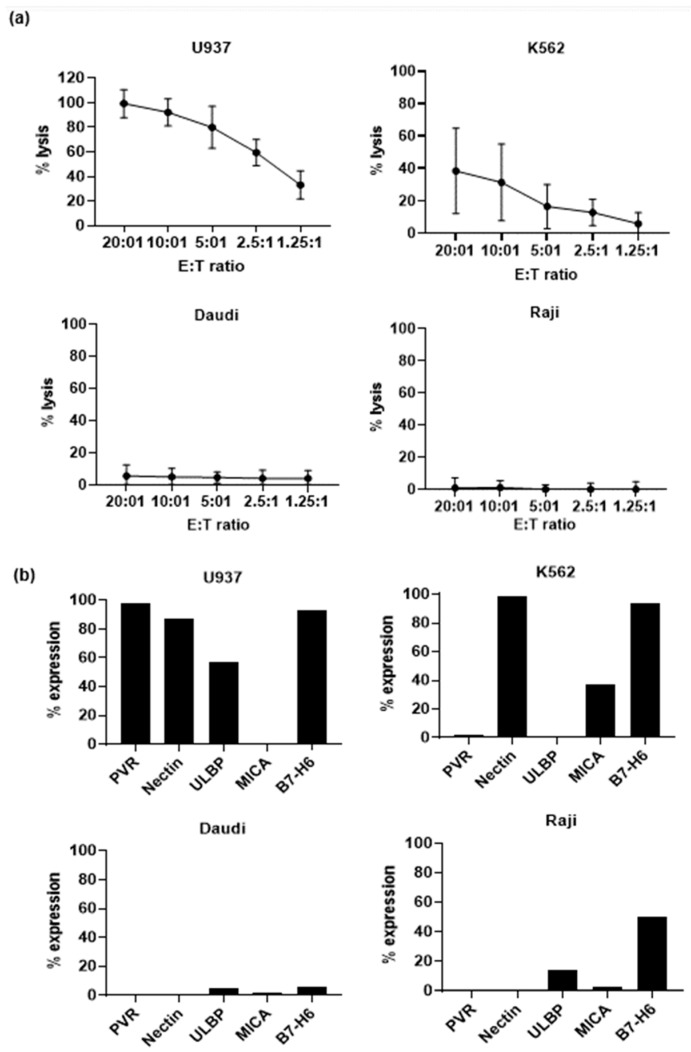
Expression of activating receptor ligands in target cells and Vδ1+ T-cell cytotoxicity. (**a**) Cytotoxic assays using cultured Vδ1+ T cells were performed against U937, K562, Daudi, and Raji cells in vitro (*n* = 6). (**b**) Expression of activating receptor ligands on target cell lines analyzed by flow cytometry and various E:T ratios (from 20:1 to 1.25:1). γδ T cells were co-cultured with calcein AM-labeled target cells at the indicated E:T ratios for 4 h. The supernatants were harvested, and fluorescence analyzed with a microplate reader. The percentage of specific lysis was calculated as [(experimental emission − spontaneous emission) / (maximum emission − spontaneous emission)] × 100. Data shown in (**a**) are the mean ± SD of % lysis of indicated tumor by Vδ1+ T cells co-cultured in 3 technical replicates for the indicated E:T ratios and from 6 independent donors.

**Figure 5 vaccines-11-00406-f005:**
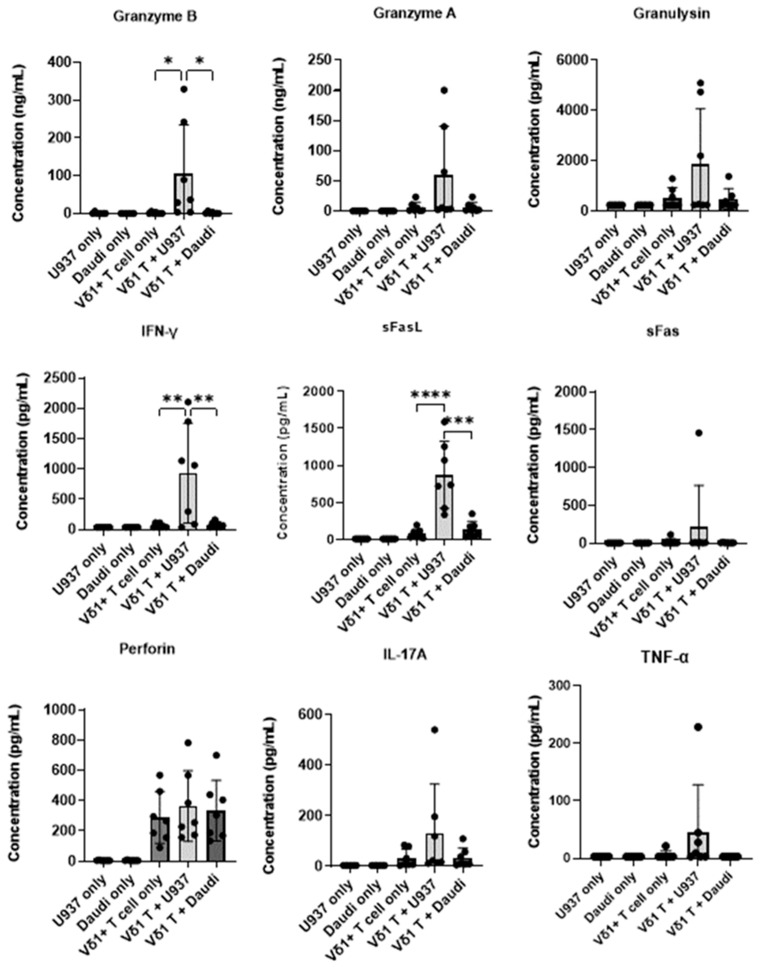
Vδ1+ T-cell secretome when co-cultured with tumor target cells. Vδ1+ T cells were co-cultured with or without target cell lines at an E:T ratio of 2:1 for 24 h. The supernatants were harvested, and cytokine profiles analyzed using a LEGENDplex human CD8/NK panel according to the manufacturer’s protocol. Experiments were analyzed with one-way ANOVA (*n* = 7). Data shown are the mean ± SD of concentration of molecules or cytokines secreted by Vδ1+ T cells from 7 cord blood donors cultured alone or with indicated tumor cells, and representative of 2 independent experiments. (* *p* < 0.05, ** *p* < 0.005, *** *p* < 0.0005, **** *p* < 0.00005). E:T, effector:target; SD, standard deviation.

**Figure 6 vaccines-11-00406-f006:**
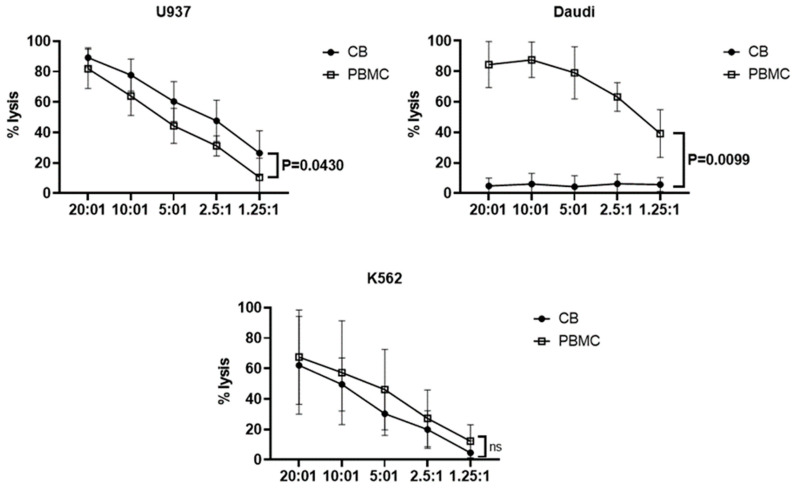
Comparison of cytotoxic effects between cord blood Vδ1+ and adult Vδ2+ T cells. For the cytotoxicity assay, Vδ1+ T cells from cord blood (*n* = 3) and Vδ2+ γδ T cells from PBMC (*n* = 3) were co-cultured with calcein AM-labeled U937, K562, and Raji cells at the indicated E:T ratios for 4 h. Supernatants were harvested, and fluorescence analyzed with a microplate reader. The percentage of specific release was calculated as [(Experimental Release − Spontaneous Release) / (Maximum Release − Spontaneous Release)] × 100. Graphs show mean ± SD of data from three independent donors. Experiments were performed in triplicate using two-way ANOVA. ns: not significant.

## Data Availability

The data presented in this study are available within the article.

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
