# Peer review of "In Vitro Expansion of Vδ1+ T Cells from Cord Blood by Using Artificial Antigen-Presenting Cells and Anti-CD3 Antibody"

_vaccines, 2023, doi:10.3390/vaccines11020406_

Round 1

Reviewer 1 Report

Summary

In this study, the authors determined the optimal conditions for highly efficacious expansion of Vd1+ T cells from cord blood (CB), achieving purity of > 90%. They found that activation of CD3 signaling was important in promoting preferential expansion of CB Vd1+ T cells and further demonstrated that the in vitro cytotoxicity of CB Vd1+ T cells against tumor cells correlated with the expression of DNAM-1 ligands PVR and Nectin on the surface of the latter cells. Expanded Vd1+ cells targeted various tumor cells and produced cytotoxic cytokines. This reviewer appreciates that such a workflow to enhance generation of CB Vd1+ cells could potentially be applied in the clinic. Key references related to expansion of gd T cell subsets were cited in the article. Although language could be improved, the article is largely intelligible to a scientifically trained reader and the authors’ conclusions were substantiated by data generated from sound experimental design. Nonetheless, there are concerns which will have to be addressed before the manuscript can be further considered for publication.

General / major comments

·      The authors showed good correlation between in vitro cytotoxicity of expanded cord blood Vd1+ T cells and expression of key activating receptors, in vivo data will strengthen this conclusion.

·      What is/are the mechanism(s) underlying increased sensitivity of Vd1+ compared with Vd2+ T cells to CD3 stimulation? Understanding such mechanisms will contribute to improvements in protocols that preferentially expand Vd1+ T cells ultimately for clinical use in tumor immunotherapy.

·      It is known that anti-CD3 activation promotes apoptosis of both ab and gdT cells. Did the authors observe increased apoptosis of cord blood Vd1+ T cells which were activated by anti-CD3 Ab despite enhanced proliferation? Did co-culture of Vd1+ T cells with artificial antigen presenting cells (aAPCs) rescue the apoptotic effects of CD3 signaling?

·      The authors concluded that in vitro cytotoxicity of expanded cord blood Vd1+ T cells positively correlated with their expression of key activating receptors such as PVR and Nectin. To demonstrate that these receptors are indeed important to mediate cytotoxicity, the authors should consider knocking down or knocking out at least PVR and Nectin in Vd1+ T cells to assess if this will substantially reduce or abrogate their lysis of U937 compared with Daudi cells.

·      After the authors determined the optimal conditions that robustly expand cord blood Vd1+ T cells, they carried out experiments to characterize the expression of activating receptors, differentiation status and cytotoxic efficacy of these cells without comparison to cord blood Vd2+ counterparts except for comparison of cytotoxicity of cord blood Vd1+ T cells with adult blood Vd2+ T cells in Figure 6. This reviewer suggests that in this manuscript, in order to minimize biological differences due to cell source, the authors consider comparing the above characteristics of cord blood Vd1+ T cells with those of cord blood Vd2+ T cells. The reviewer appreciates that the authors proposed as future study in Discussion section, p12 lines 338-340 to compare the characteristics of cord with adult blood Vd1+ T cells which will address the question of whether Vd1+ T cells derived from different tissue sources are biologically similar.

Specific / minor comments

·      Abstract

1.      Typo error in “… Daudi and Raji pcells were not susceptible to Vd1+ T cells” should be corrected to “Daudi and Raji pcells were not susceptible to Vd1+ T cells”.

2.      “… culturing cord blood Vd1+ T cells by observing that Vd1+ T cells were more sensitive to CD3 signals than other subtypes of cord blood”. Authors should clarify what is meant by “other subtypes of cord blood”. Do they mean “other subtypes of gd T cells in cord blood”?

·      Pg 2 line 45

“… the peripheral T-cell pool 13(13)”. Please correct citation “13(13)”.

·      Pg 2 line 47

For clarity, suggest to rephrase “Meanwhile, for applicable expansion protocols for non-Vd2+ T cells, agonistic antibodies stimulating gd T cells, artificial antigen presenting cells (aAPCs) have been successfully used in expanding gd T cells” to “Meanwhile, agonistic antibodies stimulating gd T cells and artificial antigen presenting cells (aAPCs) have been successfully used in expansion protocols for non-Vd2+ T cells”.

·      Figure 1 legend

The authors stated 31 blood cord donors were assessed for their frequencies of TCR Vd1+, TCR Vd2+ and TCR Vd1-Vd2- cells in scatter plot (a). They should also state how many donors were assessed for their frequencies of TCR Vd populations in each graph of (b) and (c) and whether standard error of the mean (SEM) or standard deviation (SD) was used to indicate error bars: “Data shown in (b) and (c) are the mean ± SEM (or SD?) of Vd T cell frequencies from ? independent cord blood donors”. Furthermore, the use of irradiated K562-based aAPCs expressing costimulatory molecules with or without anti-CD3 Ab for what duration should be mentioned in (b) and (c).

·      Figure 1(a) and 1(c)

Authors are advised to show representative flow cytometry plots of TCR Vd1+, TCR Vd2+ and TCR Vd1-Vd2- cells from three cord blood donors in addition to the scatter (Figure 1(a)) or bar plots (Figure 1(c)) either in main Figure 1 or supplementary figure. This will allow the reader to visualize how the different TCR Vd populations were gated. In labels on horizontal axes, authors should specify % of ? T cells, e.g. % of total Vd1+ T cells.

·      Figure 1(b)

Authors should describe clearly in Materials and Methods section 2.2 how various Vd T cells were enumerated and fold expansions were calculated. What was used as baseline cell number to calculate fold expansion?

·      Figure 1(b) and (c)

In labels on horizontal axes, replace “anti-CD3-” with “media alone” or “empty vehicle” and “anti-CD3+” with “anti-CD3”. Current labels are not conventional.

·      Section 3.1 p4, line 173

Suggest to replace “we established a culture method of pure Vd1+ T cells” with “we established a culture method to expand pure Vd1+ T cells”.

·      Figure 2 legend

Similar to Figure 1, authors should state how many donors were assessed for their frequencies of TCR Vd populations in each graph of (b) and (c) and whether SEM or SD was used: “Data shown in (a) are representative of Vd T cell frequencies from ? cord blood donors. Data shown in (b) and (c) are the mean ± SEM (or SD?) of Vd T cell frequencies from 7 independent cord blood donors”. The use of irradiated K562-based aAPCs expressing costimulatory molecules with anti-CD3 Ab for what duration should also be mentioned.

·      Figure 2 legend

Was statistical comparison between graph bars “7 days after sorting” and “14 days sorted Vd1+ T cells” performed?

·      Section 3.2 p5, line 187

The authors stated that Vd1+ T cells sorted on day 7 were “cultured for two more weeks”. The authors examined expression of activating/inhibitory surface molecules and differentiation status of sorted cells after one additional week of culture, i.e. day 14. Did the authors analyse the cells on day 21? How do frequencies of central memory (CD45RA-CD27+) and effector memory (CD45RA-CD27-) Vd1+ T cells compare with corresponding frequencies on day 14?

·      Figure 3

In labels on horizontal axes, authors should specify “% of total Vd1+ T cells” and not merely “% expression”.

·      Section 3.3

p7, line 233

“… U937 cells showed the highest susceptibility to Vd1+ T cells” should be rewritten more clearly as to “… U937 cells showed the highest susceptibility to killing by Vd1+ T cells”. Subsequent instances of similar statements should be corrected.

p8, lines 241-244

Please provide details of the correlation analyses that were carried out.

·      Figure 4 legend

Typo error for “Figureure 4” to be corrected. (a) Authors should specify the number of replicates used for each E:T ratio in cytotoxicity assays and number of independent experiments performed for each tumor cell line: “Data shown in (a) are the mean ± SEM (or SD?) of % lysis of indicated tumor by Vd1+ T cells co-cultured in ? technical replicates for the indicated E:T ratios and representative of ? independent experiments”. Horizontal axis of each lysis graph to be labelled clearly with “E:T ratio”. (b) Please specify which ULBP ligands were examined and include description of the Ab(s) used to stain the ligands in Materials and methods. Since bar graphs are used, did the authors analyse mean fluorescence intensities (MFIs) as measures of expression of the various activating receptor ligands? If indeed % expression instead of MFI was used, authors are advised to show representative flow cytometry plots to explain how % expression was obtained. Authors should specify data are representative of how many independent experiments.

·      Figure 5

Similar to Figure 4, authors should specify: ““Data shown are the mean ± SEM (or SD?) of concentration of molecules or cytokines secreted by Vd1+ T cells from 7? cord blood donors cultured alone or with indicated tumor cells and representative of ? independent experiments”.

·      The authors observed that the levels of cytotoxicity and associated secretome of expanded cord blood Vd1+ T cells against tumor cells in vitro correlated positively with expression of activating ligands by the latter cells. Did the authors assess the capacity of expanded Vd1+ T cells to suppress selected tumors engrafted in immunodeficient mice and examine if this also correlated with the expression of activating ligands by Vd1+ T cells isolated from blood or tumors? This would

·      TNF-a is another major cytokine secreted by gd T cells in response to stimulation by tumor cells as mentioned by the authors in Discussion section. Were the authors able to detect TNF-a secretion by Vd1+ T cells upon co-culture with U937 and Daudi cells?

Reviewer 2 Report

In this manuscript by Hur et al, authors aim to analyze the effect of CD3 stimulation on Vd1 cells from cord blood. Authors only compared one expansion condition with or without CD3 antibody and therefore, the conclusion that “γδ T cells proliferated only in the presence of CD3 (line 296)” does not seem appropriate. The manuscript is not well organized and at times it is very difficult to follow the figures/results due to the name provided to the conditions. Further experiments/analysis might be required to demonstrate lack of Daudi cell killing by Vd1 cells, which contrasts with previous studies. Figure legends are not properly explained most of the time and number of donors/replicates used are not obvious. Statistical analysis should be reviewed with a biostatistician.

Figure 1:

What does percent mean? Percent of total? CD3?..

Please, show a representative flow cytometry plot

Did the absolute numbers also increase? Absolute numbers should be shown within the figure

The legend should clearly state number of participants. Were APCs used in the experiment?

A)     Break the axis to clearly see frequencies of Vd1 and Vd2 populations.

B)     Why there are only 5 data points? What do they represent? Show both pan-gd T cell fold expansion and break it into Vd1 and Vd2 T cell subsets.

C)     Show the graph as individual data points (similar to Figure 1B), how many donors were included?

Since the data does not follow a normal distribution, paired non-parametric tests seems more appropriate.

Showing actual p values rather than < than is more appropriate.

Figure 2:

Authors need to show absolute numbers. Vd2 cells seem to be dying, even in the presence of IL-2, but that might be due to the overall frequency.

The terms chosen: “7 days before sorting” “7 days after sorting”…are confusing. Authors should use terms related to culturing/expansion conditions or similar, or provide such clarification in the legend.

Figure 3:

What was the phenotype before expansion and after sorting? These results should be shown in the figure.

Representative flow plots should include the modulation of the phenotypes during the different expansion stages: from baseline to culturing the presence of APCs to post-sorted Vd1 cells.

“Effector cells” should be defined by the convention TEMRA.

Individual data should be presented for Figure 3B. Inhibitory markers should be shown as Figure 3C. For inhibition, scaling the y axis down to around 20, will allow to clearly see the level of expression of PD1 CTLA4. Showing the modulation of the receptors would also help clarify why PD1 expression is low after stimulation. It is possible that FACS-sorting has skewed the populations? Authors refer to CD8 T cell differentiation (line 227), what about PD1/CTLA-4 in GD cells? i.e., PMID: 21268005

Results shown here do not confirm previous studies demonstrating Daudi cell killing by Vd1 cells. Can authors discuss the discrepancies with other studies? Since the chosen assay is rather non-specific, further flow cytometry assays using a viability dye to show Daudi cell killing are required.

Explanation of Figure 4  in the main text is missing, only the legend is there. Legend is also confusing. Where is the phenotype of Vd1 cells? What ratio did authors use for the cell line phenotyping?

How was the viability of the cell lines in figure 5? Individual data of the 7 donors must be represented.

Figure 6: unclear how many donors were used. Were three technical or biological replicates used?

Round 2

Reviewer 2 Report

Hur et al have partially improved the manuscript. There are still many unclear points and issues that must be resolved.

Still unclear why authors make the comparison between cord blood Vd1 and adult peripheral blood. At least include a more thorough discussion of what they find in this study with what has been published about expanded adult peripheral Vd1s. The only part of the discussion that really dives into that is when they discuss NKp30. Inclusion of more discussion about expanded peripheral Vd1 cells and killing of these cell lines is required.

CD45RA+ CD27- cells are called by convention TEMRA. Not sure why authors do not term them by the convention.

Lines 270-272: Showing three technical replicates rather than the six independent experiments do not provide sufficient support to the conclusions. Figure 4b needs to show how many donors and error bars and statistics.

Figure 4 and supplemental figure 6: Regrettably, the flow analysis used to generate this data is not appropriate. Gates seem to be placed randomly. Where FMOs or other controls used? Please, include those. In addition, the expression of all these markers does not follow a bimodal distribution and therefore, the analysis should  be performed using MFI to compare among the different cell lines. Why is Raji cell line showing two populations? What is red and what is blue?

Authors didn't characterize or really talk about the properties of expanded Vd2s used for comparisons in Figure 6. We know they can also upregulate NKG2D and DNAM-1. They also make these definitive statements in the intro and discussion such as "Vδ2+ T-cell activation is dependent on target cell exposure/accumulation of HMBPP and IPP." which should be amended to include papers showing Vd2s can also be activated or exert cytotoxicity through recognition of non-TCR ligands expressed by target cells. 

Previous studies have used Daudi cells to show Vd1 killing (Fisch et al, Zhou et al, Bukowski, Rothenfusser…) 

Round 3

Reviewer 2 Report

Authors have responded to my questions and improved the manuscript. One last point needs to be addressed.

Some papers with Daudi cell killing by Vd1 cells that should be discussed in the manuscript

  • PMID: 2185329
  • PMID: 21666706
  • PMID: 12600753

Author Response

Authors have responded to my questions and improved the manuscript. One last point needs to be addressed.

Some papers with Daudi cell killing by Vd1 cells that should be discussed in the manuscript

  • PMID: 2185329
  • PMID: 21666706
  • PMID: 12600753

Answer:

I sincerely thank you for providing literature that can complement our limited knowledge.

I read the three papers presented and added the discussion on Daudi cell killing by Vδ1+ T cells to the discussion section as follows.

However, I didn't cite a paper in 1990 (J. Exp. MED.) because the distinction between Vδ1 and Vδ2 subsets was not clear.

“Cord blood Vδ1+ T cells showed high cytotoxicity in K562 and U937 cells, but not in Raji or Daudi cells. The anti-γδ TCR antibody could expand both the Vδ1 and Vδ2 subsets of peripheral blood γδ T cells. In the comparison of the cytotoxic activity between them, purified each Vδ1 and Vδ2 subset exhibited similar cytotoxicity against Daudi and K562 cells [36]. However, a Vδ1+ T cell line established from human peripheral blood by immortalization with Herpesvirus saimiri was able to specifically recognize K562 cells, but Daudi cells were resistant to Vδ1+ T cell killing [37]. As such, peripheral blood-derived Vδ1+ T cells show different degrees of cytotoxicity to Daudi cells depending on the culture conditions, so it is necessary to confirm using Vδ1+ T cells amplified under the same culture conditions as in this study in the future.”